# Synthesis and Pharmacological Evaluation of Hybrids Targeting Opioid and Neurokinin Receptors

**DOI:** 10.3390/molecules24244460

**Published:** 2019-12-05

**Authors:** Karol Wtorek, Anna Adamska-Bartłomiejczyk, Justyna Piekielna-Ciesielska, Federica Ferrari, Chiara Ruzza, Alicja Kluczyk, Joanna Piasecka-Zelga, Girolamo Calo’, Anna Janecka

**Affiliations:** 1Department of Biomolecular Chemistry, Medical University of Lodz, Mazowiecka 6/8, 92-215 Lodz, Poland; karol.wtorek@umed.lodz.pl (K.W.); anna.adamska@umed.lodz.pl (A.A.-B.); justyna.piekielna@umed.lodz.pl (J.P.-C.); 2Department of Medical Sciences, Section of Pharmacology, University of Ferrara, 44121 Ferrara, Italy; frrfrc2@unife.it (F.F.); rzzchr@unife.it (C.R.); clg@unife.it (G.C.); 3Faculty of Chemistry, University of Wroclaw, 50-383 Wroclaw, Poland; alicja.kluczyk@chem.uni.wroc.pl; 4Institute of Occupational Medicine, Research Laboratory for Medicine and Veterinary Products in the GMP Head of Research Laboratory for Medicine and Veterinary Products, 91-348 Lodz, Poland; Joanna.Zelga@imp.lodz.pl

**Keywords:** opioid receptors, neurokinin-1 receptor, peptide synthesis, receptor binding studies, functional assay, writhing test, tolerance

## Abstract

Morphine, which acts through opioid receptors, is one of the most efficient analgesics for the alleviation of severe pain. However, its usefulness is limited by serious side effects, including analgesic tolerance, constipation, and dependence liability. The growing awareness that multifunctional ligands which simultaneously activate two or more targets may produce a more desirable drug profile than selectively targeted compounds has created an opportunity for a new approach to developing more effective medications. Here, in order to better understand the role of the neurokinin system in opioid-induced antinociception, we report the synthesis, structure–activity relationship, and pharmacological characterization of a series of hybrids combining opioid pharmacophores with either substance P (SP) fragments or neurokinin receptor (NK1) antagonist fragments. On the bases of the in vitro biological activities of the hybrids, two analogs, opioid agonist/NK1 antagonist Tyr-[d-Lys-Phe-Phe-Asp]-Asn-d-Trp-Phe-d-Trp-Leu-Nle-NH_2_ (**2**) and opioid agonist/NK1 agonist Tyr-[d-Lys-Phe-Phe-Asp]-Gln-Phe-Phe-Gly-Leu-Met-NH_2_ (**4**), were selected for in vivo tests. In the writhing test, both hybrids showed significant an antinociceptive effect in mice, while neither of them triggered the development of tolerance, nor did they produce constipation. No statistically significant differences in in vivo activity profiles were observed between opioid/NK1 agonist and opioid/NK1 antagonist hybrids.

## 1. Introduction

Due to their role in pain perception and modulation, opioid receptors (µ, δ, and κ, or MOR, DOR, and KOR, respectively) are very important targets in medicinal chemistry. The plant alkaloid morphine and its derivatives, which elicit their analgesic effect mostly through the activation of MOR [1], are often the only choice for the management of severe pain [2]. However, the long-term use of these drugs in chronic pain states causes the development of tolerance, which in turn necessitates dose escalation [3]. As a result, the development of side effects, including the inhibition of gastrointestinal transit, respiratory depression, and physical dependence, occurs [4]. Therefore, the dissociation of analgesia from the adverse side effects elicited by MOR agonists is the main goal in the search for better and safer analgesics.

In the past decade the efforts of chemists in synthesizing new opioid analogs have been concentrated on obtaining multifunctional opioid ligands, interacting simultaneously with more than one opioid receptor type [5,6,7]. For example, compounds with a MOR agonist/DOR antagonist profile showed fewer side effects and enhanced efficacy [8].

Since opioid peptides are not the only modulators of pain signals in the central nervous system (CNS), a new approach in the search for more efficient analgesics with limited side effects is to combine opioids with other neurotransmitters involved in pain perception (e.g., cholecystokinin, neurotensin, substance P, etc.) [9,10,11,12]. Such novel chimeras, also known as multitarget ligands, may interact independently with their respective receptors and potentially produce more effective antinociception [13].

The tachykinin undecapeptide substance P (SP: Arg-Pro-Lys-Pro-Gln-Gln-Phe-Phe-Gly-Leu-Met-NH_2_) is a neurotransmitter/neuromodulator which is known to transmit pain signaling from the periphery to the CNS. SP acts through the activation of neurokinin 1 receptor (NK1), which is found in both the central and peripheral nervous systems and is associated with pain responses related to noxious stimuli [14]. Antagonism at the NK1 blocks the signals induced by SP and can inhibit the enhanced secretion of SP and increased expression of NK1 in prolonged pain states. That makes NK1 antagonists potential therapeutic agents for pain relief [15,16,17].

The pharmacological blockade of NK1 with an antagonist seems logical in the design of hybrid peptides with improved activity. The rationale for the synthesis of opioid/NK1 antagonist hybrids is also supported by the documented co-localization of opioids and NK1 in nervous structures in the transmission of nociceptive impulses [12].

Several hybrid opioid/NK1 antagonists have been reported so far. As the opioid part, fragments or analogs of [Met]enkephalin [18], biphalin [19], dermorphin [20], or other opioids [21,22] have been used and connected with NK1 antagonists. One example of such a MOR agonist/NK1 antagonist is the compound Dmt-d-Arg-Aba-Gly-*N*-methyl-*N*-3′,5′-di(trifluoromethyl)benzyl, which produced potent analgesic effects upon chronic administration but still manifested a tolerance profile similar to that of morphine [23].

In contrast to the hyperalgesic effects of SP, it was demonstrated that in low doses, SP can intensify opioid-mediated analgesia in a naloxone-reversible manner, probably by triggering endogenous opioid peptide release [24]. Linking the *C*-terminal SP fragments with morphine or opioid peptides gave hybrids which produced a strong analgesic effect, with low or no tendency to develop opioid tolerance following central administration to rats [25,26,27,28].

Foran et al. [25] described the endomorphin-2 (EM-2)/SP-7-11 chimera (ESP7) with overlapping Phe-Phe residues, which produced opioid-dependent antinociception without loss of potency over a five-day period, suggesting that co-activation of MOR and NK1 is essential for maintaining opioid responsiveness.

Kream et al. [26] synthesized a hybrid of morphine covalently conjugated through a succinic acid linker to SP_3-11_ (designated MSP9). This analog was shown to activate MOR and KOR, as well as NK1, and its antinociceptive effect most likely depended on the potent functional coupling of NK1 with both opioid receptors.

Small peptides are in general not suitable as drugs, since they are metabolically unstable and often degrade in a few minutes. Among other strategies, cyclization has turned out to be a useful tool for generating analogs with enhanced chemical and enzymatic stability, as well as improved pharmacodynamic properties [29]. In the last several years, we focused our research on the synthesis of cyclic analogs based on the sequence of endomorphin-2 (EM-2; Tyr-Pro-Phe-Phe-NH_2_), with incorporated bifunctional amino acids that make ring closure possible. The cyclic peptide Tyr-c[d-Lys-Phe-Phe-Asp]NH_2_ (**1**) displayed a mixed MOR/KOR affinity profile, enzymatic stability, and strong and long-lasting antinociceptive activity after either intracerebroventricular (icv) or peripheral administration, which indicated its ability to cross the blood–brain barrier (BBB) [30,31].

Here, we report the synthesis and pharmacological evaluation of hybrids combining this cyclopeptide with either the SP pharmacophore or NK1 antagonist (spantide II) fragments in order to compare the antinociceptive potential of these two types of hybrids and to enhance our knowledge on the cross-talk between opioid and neurokinin systems in pain perception and regulation.

The largest obstacle in joining two molecules is possible interference between them, which may lead to partial or even complete loss of affinity at the respective receptors. In this report, the binding and activation profiles of the hybrid analogs at the opioid and NK1 receptors were investigated, followed by in vivo studies of antinociceptive activity and tolerance development in mice.

## 2. Results

### 2.1. Chemistry

The structures of the new hybrids are presented in Figure 1. The peptides were synthesized by the solid-phase procedure using Fmoc/*t*Bu chemistry with the hyper-acid labile Mtt/*O*-2 Ph*i*Pr groups for selective protection of amine/carboxyl side-chain groups engaged in the formation of the cyclic fragment. TBTU (2-(1H-benzotriazole-1-yl)-1,1,3,3-tetramethylaminium tetrafluoroborate) was used for coupling reactions. All compounds were purified by semipreparative RP HPLC (reverse phase high-performance liquid chromatography), and their identity was confirmed by high-resolution mass spectrometry (ESI-HRMS). The purity of the compounds characterized by analytical RP HPLC was determined to be ≥95%. The detailed analytical data of the synthesized peptides are provided in the Appendix A.

### 2.2. Receptor Binding Affinity

A radioligand binding assay was performed to determine the opioid receptor (OR) binding affinities of the novel analogs using commercially available membranes of Chinese hamster ovary cells (CHO) transfected with human recombinant ORs. [^3^H]DAMGO, [^3^H]deltorphin-2, and [^3^H]U-69593 were used as the radioligands for MOR, DOR, and KOR, respectively. The results are summarized in Table 1. Tyr-[d-Lys-Phe-Phe-Asp]NH_2_ (**1**) was used as a reference opioid compound. The novel hybrids, with the exception of **3,** showed MOR affinity in the nanomolar range. None of them acquired significant DOR affinity. At the KOR, analog **7**, containing the shortest SP fragment, exhibited the strongest binding.

The binding assay was also used to examine the affinity of the synthesized hybrids for NK1 in commercial membranes of CHO cells stably expressing human NK1. [^3^H][Sar^9^, Met(O_2_)^11^]SP was used as a competing radioligand. SP was included as a parent agonist (Table 1).

### 2.3. Calcium Mobilization Functional Assay

The functional activity of the hybrids was evaluated at all three ORs in calcium mobilization assay in which CHO cells co-expressing human recombinant opioid receptors and chimeric G proteins were used to monitor changes in intracellular calcium levels [32,33]. These changes reflect activation of the G-protein-coupled receptors (GPCR) and can be used for the pharmacological characterization of novel agonist and antagonist ligands [34,35].

The concentration–response curves were obtained for all hybrids (Appendix A). The agonist potencies (pEC_50_) and efficacies (α) of the tested ligands are summarized in Table 2.

At the MOR, the parent analog **1** showed full efficacy and potency (pEC_50_ = 8.98, α = 0.98), even higher than EM-2 (pEC_50_ = 8.22, α = 1.00). Analogs **2** and **3** containing NK1 antagonist fragments showed lower efficacy (α = 0.66 and 0.72, respectively) and much lower potency (30- and 100-fold, respectively) as compared with **1**. For hybrids **4**–**7**, containing hexa-, penta-, tetra-, and tri-peptide *C*-terminal fragments of SP, activation of MOR depended on the length of the SP fragment, with the highest pEC_50_ value observed for analog **7**, containing the shortest SP sequence (Gly-Leu-Met-NH_2_).

Consistent with the binding results, analogs **1–4** were able to increase intracellular calcium levels at DOR only at the highest concentration tested, and peptides **5–7** showed potency an order of magnitude lower than that of [D-Pen^2,5^]enkephalin (DPDPE), used as a reference DOR ligand. At the KOR, the cyclic opioid **1,** hybrids **2** and **3** containing NK1 antagonist fragments, and **6** and **7** with SP fragments all stimulated calcium release with high potency and efficacy. Especially, analog **7** mimicked the stimulatory effect of dynorphin A, showing maximal effect and a similar value of potency (pEC_50_
**=** 8.80).

The calcium mobilization assay results were well correlated with the results of the binding assays.

Calcium mobilization studies were also performed for all new hybrids using CHO cells expressing the human NK1 (CHO_NK1_). In these cells, SP, used as a standard, stimulated intracellular calcium mobilization in a concentration-dependent manner with high maximal effect and potency (pEC_50_ = 9.08). Analog **4** mimicked the stimulatory effects of SP with a fourfold lower potency value (pEC_50_ = 8.45), while all other hybrids were much less active, with potency decreasing in the same order as SP fragment length. These data are summarized in Table 3.

Compounds 2 and 3, which were inactive as agonists, were then tested as antagonists in inhibition response experiments against SP, and the effects were compared with an NK1 antagonist, aprepitant. Aprepitant tested as an agonist up to 10 µM did not modify per se the intracellular calcium levels in the CHONK1 cells. In inhibition response experiments, increasing concentrations of aprepitant and analogs 2 and 3 (0.01 nM to 10 µM) were tested against 10 nM SP. All three compounds inhibited the effect of SP in a concentration-dependent manner (Figure 2). The pKB values of these compounds are summarized in Table 4.

### 2.4. Antinociceptive Activity

For the assessment of antinociceptive activity, two hybrid analogs were chosen, one with an opioid agonist/NK1 antagonist profile (**2**) and one with an opioid agonist/NK1 agonist domain (**4**). For comparison, cyclic opioid analog **1** was included in the study as a positive control. This analog was shown in our previous paper [30] to exert a very strong antinociceptive effect in the hot-plate test, much stronger than EM-2. The writhing test was used as an acute pain model. This test involves intraperitoneal (i.p.) injection of acetic acid which results in abdominal constriction, causing the mice to writhe. Peptides or saline (control) were administered i.p. over a concentration range of 0.3 to 5 mg/kg at 15 min before the injection of acetic acid (0.5%, 10 mL/kg). The baseline number of writhes in the saline-treated animals was about 35. The i.p. administration of peptides to mice not treated with acetic acid did not evoke any writhes (data not shown). All three tested analogs (**1**, **2**, and **4**) significantly decreased the number of writhes (down to averages of 6.4, 12.4, and 4.5, respectively) at the dose of 5 mg/kg, and the effect was dose-dependent (Figure 3).

### 2.5. Tolerance

To examine whether mice developed tolerance to hybrids **2** and **4**, these analogs and analog **1** for comparison were injected once daily (5 mg/kg) for seven consecutive days. After repeated administration, the antinociceptive effect was determined on Day 7, performing the writhing test as above. The analgesic activity of opioid analog **1** was drastically reduced (number of writhes increased from 6 to 17). In contrast, on the seventh day of administration, hybrids **2** and **4** still produced as strong an antinociceptive effect as on Day 1, indicating that they did not cause tolerance development (Figure 4).

### 2.6. Stool Mass and Consumption of Food and Water

In order to determine the influence of prolonged administration of the tested compounds on gastrointestinal passage, stool was collected for six days from mice used in the tolerance development test. Opioid analog **1**, used as a positive control, was shown to cause serious constipation in mice compared to the control group. For both hybrids **2** and **4**, no statistically significant differences in stool mass were observed.

Animals injected daily with saline or the tested compounds were also used to determine the influence of analogs on food and water consumption. Stool water content and food and water consumption were unchanged in all tested groups compared with the control (Figure 5).

## 3. Discussion

Opioid and NK1 receptors are highly expressed in the CNS [36], and both play important roles in the direct and indirect control of pain signal transmission and modulation [37].

Here, we report on the synthesis and the initial pharmacological testing of several hybrid peptides with opioid agonist/NK1 antagonist and opioid agonist/NK1 agonist profiles. We performed structure affinity/activity relationship studies, seeking hybrids that could simultaneously and potently bind opioid and NK1 receptors. Both types of hybrids were designed with the same opioid fragment to make the comparison of their activity easier. It was already documented in the literature that simultaneous activation of both MOR and KOR intensifies the analgesic effect of pure MOR agonists [38]. Therefore, for the construction of the hybrids, opioid peptide **1**, displaying mixed MOR/KOR affinity, very good enzymatic stability, and antinociceptive activity even stronger than that of MOR-selective EM-2, was chosen [30]. This cyclic opioid was linked with either NK1 antagonist or agonist fragments of various length. As NK1 antagonists, *C*-terminal hexa- or pentapeptide portions of spantide II were used. NK1 agonists were represented by hexa-, penta-, tetra-, or tripeptide *C*-terminal fragments of SP. All new hybrids displayed reduced but still high binding affinity and agonist activity at the MOR which could be attributed to the opioid portion and which was crucial in order to achieve in vivo analgesia. The affinity for MOR of the hybrids linking opioid and SP fragments increased inversely to the length of the SP fragment. All new hybrids also retained quite high affinity for KOR, only slightly decreased in comparison to parent peptide **1**. Significant affinity for NK1 was acquired for only hybrids with penta- and hexapeptide fragments of spantide II and a hexapeptide fragment of SP. Spantide-II-containing chimeras **2** and **3** were shown to bind with quite good affinity to NK1 but were inactive in the functional test, suggesting that they could be NK1 antagonists. This assumption was confirmed in the inhibition response experiments against SP. Taken together, the obtained results showed that opioid/NK1 antagonist hybrids stimulated opioid receptors and simultaneously behaved as antagonists at NK1.

On the basis of these data, we can conclude that the attachment of SP or spantide II fragments to the opioid peptide did not prevent binding of the hybrids to MOR and KOR, and the K_i_ values were, with one exception (hybrid **3**), only 3- to 23-fold higher when compared to the parent opioid. Activation of the NK1 receptor required a hexapeptide fragment of SP.

Hybrids **2** and **4**, which could interact with both opioid and NK1 receptors, were selected for the in vivo assay, which was conducted in an acute pain model (writhing test) in mice. Despite the lower affinity of the hybrids for opioid and NK1 receptors, the analgesic effect they evoked was comparable with that of parent opioid **1.** No statistically significant differences in the in vivo activity profile were observed between opioid/NK1 agonist and opioid/NK1 antagonist hybrids bearing the same opioid fragment. None of the hybrids caused tolerance or constipation development, unlike **1**.

We can assume that hybrid **2** blocked the pronociceptive signals induced by the endogenous SP, which in turn enhanced the effect produced by the opioid part. Hybrid **4** could intensify the antinociception by triggering the release of endogenous opioids, as reported for the co-administration of morphine with small doses of SP [24]. Signaling cross-talk between opioid receptors mediating analgesia [39,40] and the NK1 receptor which is responsible for pain perception is well documented [41,42]. The co-administration of NK1 antagonists together with opioids was considered an option in chronic pain management, since NK1 blockade might be able to reduce the development of opioid tolerance, physical dependence, and withdrawal [43,44]. In preclinical animal studies, NK1 antagonists showed a promising profile, attenuating the nociceptive responses caused by inflammation or nerve damage [15,16,17,45]. However, they failed to exhibit efficacy in clinical trials [46]. On the other hand, NK1 agonists were reported to counteract the development of opioid tolerance which is linked to desensitization of the opioid receptors. The desensitization is caused by the administration of opioids, especially when it is prolonged, and results in the progressive reduction of signal transduction [47]. Interestingly, in some cases an NK1 agonist, SP, can increase the recycling and enhance the resensitization of MOR [48,49]. Moreover, it was shown that either morphine or DAMGO promoted rapid endocytosis of MOR in striatal neurons, whereas the simultaneous activation of NK1 with SP inhibited this regulatory process. Therefore, bivalent hybrids which bind simultaneously with opioid and NK1 receptors seem to be the reasonable choice for studying complicated interactions between these two systems.

Further experiments including conformational analyses and molecular docking studies will be performed to disclose the structural determinants responsible for the binding of both types of hybrids to opioid receptors.

## 4. Materials and Methods

### 4.1. General Methods

Most of the chemicals and solvents were obtained from Sigma Aldrich (Poznan, Poland). Protected amino acids were provided by Trimen Co (Lodz, Poland), and MBHA Rink-Amide peptide resin (100–200 mesh, 0.8 mmol/g) was provided by NovaBiochem. Opioid radioligands, [^3^H]DAMGO, [^3^H]deltorphin-2, and [^3^H]U-69593, and human recombinant ORs and NK1 receptor came from PerkinElmer (Krakow, Poland). GF/B glass fiber strips were purchased from Whatman (Brentford, UK). Analytical and semi-preparative RP HPLC was performed using a Waters Breeze instrument (Milford, MA, USA) with a dual absorbance detector (Waters 2487). The ESI-MS experiments were performed on a Bruker FTICR (Fourier transform ion cyclotron resonance) Apex-Qe Ultra 7 T mass spectrometer equipped with a standard ESI source. The instrument was operated in the positive ion mode and calibrated with the Tunemix™ mixture (Agilent Technologies, CA, USA). The structures of peptides were confirmed using a Shimadzu LCMS-IT-TOF (ion trap–time-of-flight) hybrid mass spectrometer with auto-tuning in the positive ion mode.

### 4.2. Peptide Synthesis

All peptide hybrids were synthesized by the standard solid-phase procedure on MBHA Rink-Amide peptide resin using the N^α^-Fmoc strategy and TBTU as a coupling reagent, according to the method described elsewhere [50]. Final products were purified by RP-HPLC on a Vydac C_18_ column (10 µm, 22 × 250 mm) at a flow rate of 2 mL/min, using as an eluent a linear gradient of 0.1% TFA in water (A) and 80% acetonitrile in water containing 0.1% TFA (B) ranging from 0% to 100% B over 25 min. The purity of the analogs was at least 95%, as determined on the basis of analytical RP HPLC (Vydac C_18_, 5 µm, 4.6 × 250 mm column, 1 mL/min over 50 min) and ESI-HRMS for exact mass determination (see also Appendix A).

### 4.3. Radioligand Binding Assays

To determine the affinity of peptide analogs to respective receptors, competition binding experiments were performed, as described in detail elsewhere [51].

Commercial membranes of CHO cells stably expressing either human ORs or NK1 receptor were used. [^3^H]DAMGO, [^3^H]deltorphin-2, and [^3^H]U-69593 were employed as the competing radioligands for MOR, DOR, and KOR, respectively, while [^3^H][Sar^9^, Met(O_2_)^11^]SP was utilized for the NK1 receptor. Membranes were incubated in a 0.5 mL volume of 50 mM Tris/HCl (pH = 7.4), 0.5% bovine serum albumin (BSA), with a number of peptidase inhibitors (bacitracin, bestatin, captopril) and various concentrations of radioligands for 2 h at 25 °C. Nonspecific binding was assessed in the presence of 10 mM naloxone for ORs or SP for NK1 receptor.

### 4.4. Cell Culture

All transfected cell lines (obtained in the Department of Medical Sciences, University of Ferrara) were maintained in culture medium consisting of Dulbecco’s MEM/HAM’S F-12 (50/50) supplemented with 10% fetal bovine serum (FBS) and streptomycin (100 µg/mL), penicillin (100 IU/mL), l-glutamine (2 mmol/L), geneticin (G418; 200 µg/mL), fungizone (1 µg/mL), and hygromycin B (100 µg/mL). Cell cultures were kept at 37 °C in 5% CO_2_ humidified air. When confluence was reached (3–4 days), cells were subcultured as required using trypsin/EDTA and used for testing.

### 4.5. Calcium Mobilization Functional Assay

Calcium mobilization assay was performed as reported in detail elsewhere [51]. For the experiments, CHO cells stably co-expressing the human MOR or KOR and the *C*-terminally modified Gα_qi5_, CHO cells co-expressing DOR and the Gα_qG66Di5_ protein, and CHO cells expressing NK1 receptor were used. Cells co-expressing ORs and the chimeric G proteins were generated as described [34]. Cells expressing NK_1_ receptor were a generous gift from the laboratory of Prof. T. Costa (ISS, Rome, It). Briefly, cells incubated for 24 h in 96-well black, clear-bottom plates were loaded with medium supplemented with probenecid (2.5 mmol/L), calcium-sensitive fluorescent dye Fluo-4 AM (3 *µ*mol/L), and pluronic acid (0.01%) and kept for 30 min at 37 °C. Following aspiration of the loading solution and a washing step, serial dilutions of peptide stock solutions were added. Fluorescence changes were measured using the FlexStation II (Molecular Device, Union City, CA, USA). The maximal change in fluorescence, expressed as the percentage over the baseline fluorescence, was used to determine the agonist response. In antagonism-type experiments, tested hybrids were injected into the wells 24 min before adding an agonist (SP).

### 4.6. Animals

All animal care and experimental procedures were performed in accordance with the Medical University of Lodz recommendations, described in the Guide for the Care and Use of Laboratory Animals, and complied with the Animal Research: Reporting of In Vivo Experiments (ARRIVE) guidelines [52]. Local Bioethical Committee approval number 61/ŁB708/2017.

Male Balb/c mice (Animal Facility of Nofer Institute of Occupational Medicine, Lodz, Poland) weighing 22–25 g were used in the study. The animals were housed under standard conditions (22 ± 1 °C, 12 h light/dark cycle) with food and water ad libitum for five days before the experiments.

### 4.7. Assessment of Antinociception by Writhing Test

Antinociceptive activity was assessed by the writhing test. Mice were divided into groups of 10 animals each. A single i.p. injection of saline or a tested compound (at doses 0.3, 1, 3, or 5 mg/kg) was followed (after 15 min) by the i.p. administration of acetic acid (0.65% in 0.9% NaCl, 10 mL/kg) [53]. After 5 min recovery, specific spontaneous behaviors characterized by elongation of the body (“writhes”) were counted for 15 min.

### 4.8. Assessment of Tolerance Development

Peptides were administered at the highest dose (5 mg/kg) used before in the writhing test. In the experiment, four new groups of animals (10 mice per group, housed in one cage) underwent repeated injections (i.p.) of saline or the tested compounds (Days 1–7). On Day 7, following saline or peptide administration, the writhing test was performed, as described above. The antinociceptive activity of a compound was compared to the activity obtained after a single injection of this compound in the corresponding dose (5 mg/kg).

Immediately after the sessions with acetic acid, mice were anesthetized via 3% isoflurane (AErrane) inhalation and euthanized by cervical dislocation.

### 4.9. Stool Collection

After the i.p. injections of saline or peptides (Days 1–6), mice were placed in a separate clean cages and stool was collected for a period of 90 min. Fecal pellets were collected immediately after expulsion to avoid evaporation and placed in closed 1.5 mL tubes. Tubes were weighed to obtain the wet weight of the stool. Then, the tubes were opened and stool was dried overnight at 65 °C and reweighed to obtain the dry weight. The stool water content was calculated from the difference between the wet and dry stool weights.

### 4.10. Food and Water Consumption

Every day at the same hour (8:00 a.m.) from Day 1 to 6, the mass of consumed food (g) and water (mL) was calculated from the difference in weights of the food and water supply at the beginning and the end of the 24 h observation period.

### 4.11. Statistical Analysis and Terminology

The pharmacological terminology adopted in this paper is consistent with International Union of Basic and Clinical Pharmacology (IUPHAR) recommendations [54]. All data are expressed as mean ± SEM from *n*-many experiments, as shown in the table and figure legends. Concentration–response curves were analyzed by nonlinear regression using Graph Pad Prism 6.0 (La Jolla, CA, USA). In displacement binding assays, the values of the inhibitory constants (K_i_) were obtained from displacement curves and calculated using the Cheng and Prusoff equation [55] {log[EC_50_/(1 + [L]/K_d_)} where EC_50_ is the concentration of the ligand that displaces 50% of the radioligand, [L] is the radioligand concentration, and K_d_ is the dissociation constant of the radioligand. In functional experiments, agonist potency was expressed as pEC_50_, which is the negative logarithm to base 10 of the agonist molar concentration that produces 50% of the maximal possible effect of that agonist. Concentration–response curves were fitted with the four-parameter logistic nonlinear regression model
(1)$$\mathrm{Effect}=baseline+\frac{E_{max}-\mathrm{baseline}}{1+10^{\left(logEC50-X\right)\cdot n}}$$
where X is the agonist concentration and n is the Hill coefficient. Ligand efficacy was expressed as intrinsic activity (α) calculated as a ratio of the peptide E_max_ to the E_max_ of the standard agonist (Appendix A). The antagonist potency for ligands in inhibition response experiments was expressed as pK_B_, which was calculated as the negative logarithm to base 10 of the K_B_ from the following Equation:
(2)$$K_{B}=\left[\frac{IC_{50}}{\left({\left[2+{\left(\frac{\left[A\right]}{EC_{50}}\right)}^{n}\right]}^{1/n}-1\right)}\right]$$
where IC_50_ is the concentration of antagonist that produces 50% inhibition of the agonist response, [A] is the concentration of agonist, EC_50_ is the concentration of agonist producing a 50% maximal response, and *n* is the slope coefficient of the concentration–response curve to the agonist [56]. All statistical analyses were performed using Graph Pad Prism 6.0 (La Jolla, CA, USA). Differences between groups were analyzed with one-way ANOVA and a post hoc multiple comparison by the Student–Newman–Keuls test. A probability level of 0.05 or smaller was used to indicate statistical significance.

## Figures and Tables

**Figure 1 molecules-24-04460-f001:**
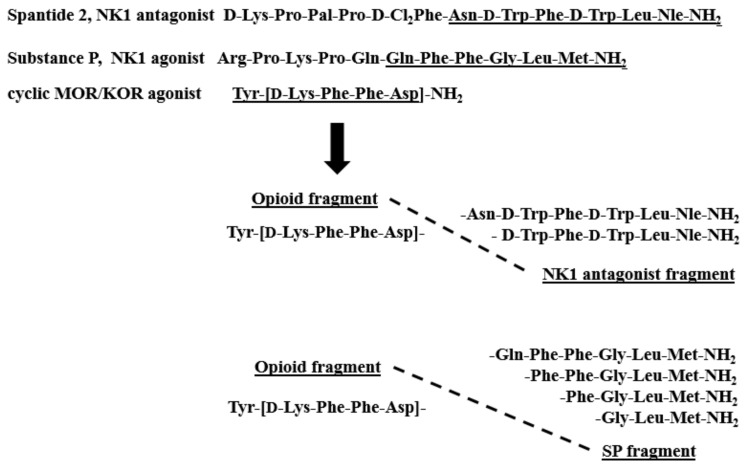
Sequences of hybrid analogs.

**Figure 2 molecules-24-04460-f002:**
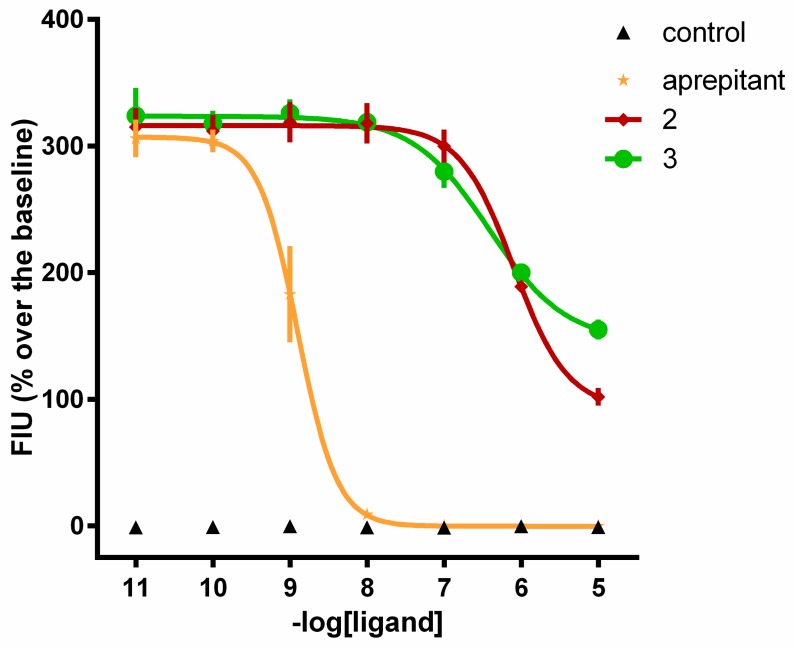
Calcium mobilization assay. Inhibition response curves to aprepitant and analogs **2** and **3** against SP (10 nM). Aprepitant alone was used as a control; *n* ≥ 3.

**Figure 3 molecules-24-04460-f003:**
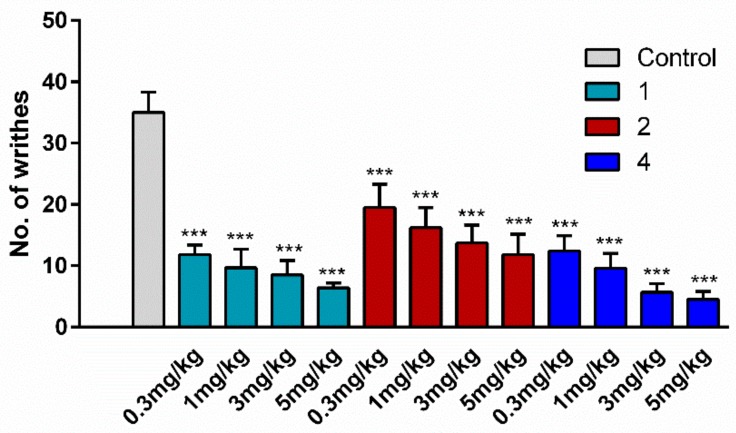
The effect of opioid analog **1** and hybrids **2** and **4** at the doses of 5, 3, 1, and 0.3 mg/kg, administered intraperitoneally (i.p.), on the number of pain-induced behaviors in mice. Peptides or vehicle were administered 15 min before the i.p. injection of acetic acid (0.5%, 10 mL/kg). The number of writhes was determined 5 min after acetic acid injection over a period of 15 min. The data represent mean ± SEM, *n* = 10. Statistical significance was assessed using one-way ANOVA and a post hoc multiple comparison by the Student–Newman–Keuls test. *** *p* < 0.001, as compared to control.

**Figure 4 molecules-24-04460-f004:**
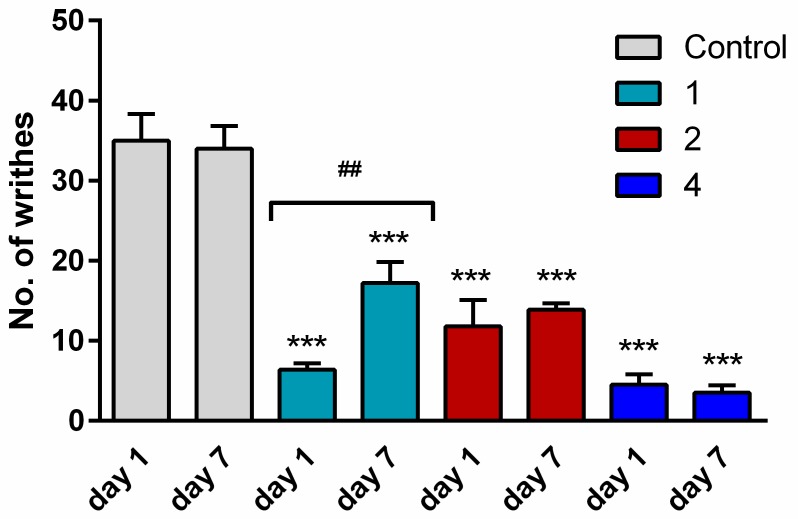
A comparison of the antinociceptive effect of single (Day 1) and repeated (Day 7) i.p. injections of opioid analog **1** and hybrids **2** and **4** at the dose of 5 mg/kg in the writhing test in mice. The data represent mean ± SEM, *n* = 10. Statistical significance was assessed using one-way ANOVA and a post hoc multiple comparison by the Student–Newman–Keuls test. *** *p* < 0.001, as compared to control.

**Figure 5 molecules-24-04460-f005:**
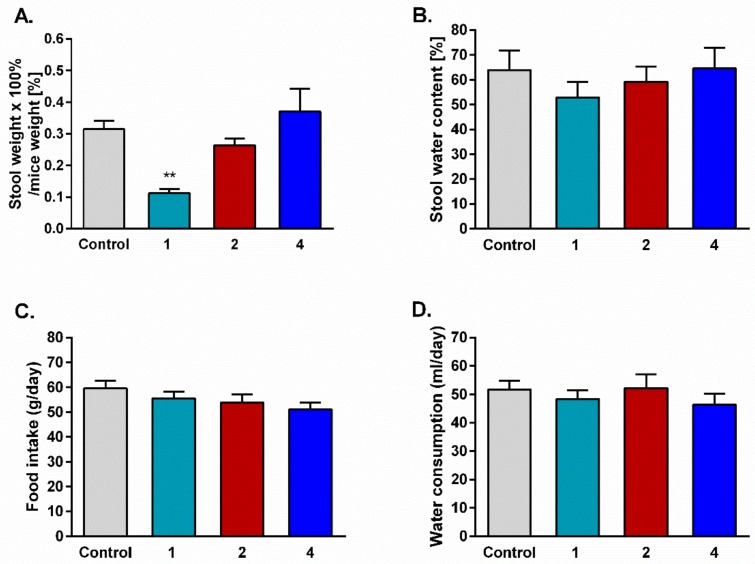
Effect of repeated i.p. injections of saline and the tested peptides on stool mass (**A**), stool water content (**B**), food intake (**C**), and water consumption (**D**). The data represent mean ± SEM, *n* = 10. Statistical significance was assessed using one-way ANOVA and a post hoc multiple comparison by the Student–Newman–Keuls test. *** *p* < 0.001, as compared to control.

**Table 1 molecules-24-04460-t001:** Receptor affinities of hybrid opioid/NK1 antagonist/agonist analogs at MOR, DOR, KOR, and NK1.

No.	Sequence	K_i_ [nM]			
		MOR ^a^	DOR ^a^	KOR ^a^	NK1 ^b^
**1**	Tyr-[d-Lys-Phe-Phe-Asp]-NH_2_	0.35 ± 0.02	170.8 ± 3.50	1.12 ± 0.20	Inactive
**2**	Tyr-[d-Lys-Phe-Phe-Asp]-Asn-D-Trp-Phe-D-Trp-Leu-Nle-NH_2_	5.99 ± 0.70	201.6 ± 2.50	7.46 ± 0.60	10.48 ± 0.60
**3**	Tyr-[d-Lys-Phe-Phe-Asp]-D-Trp-Phe-D-Trp-Leu-Nle-NH_2_	28.24 ± 1.45	212.6 ± 6.31	2.85 ± 0.44	12.82 ± 0.92
**4**	Tyr-[d-Lys-Phe-Phe-Asp]-Gln-Phe-Phe-Gly-Leu-Met-NH_2_	7.98 ± 0.97	224.8 ± 14.0	10.76 ± 0.85	15.6 ± 1.23
**5**	Tyr-[d-Lys-Phe-Phe-Asp]-Phe-Phe-Gly-Leu-Met-NH_2_	4.90 ± 0.34	96.6 ± 3.8	28.34 ± 1.71	51.6 ± 4.2
**6**	Tyr-[d-Lys-Phe-Phe-Asp]-Phe-Gly-Leu-Met-NH_2_	2.98 ± 0.14	28.7 ± 1.01	2.7 ± 0.12	2332 ± 189
**7**	Tyr-[d-Lys-Phe-Phe-Asp]-Gly-Leu-Met-NH_2_	1.14 ± 0.21	113.2 ± 4.6	0.89 ± 0.04	8128 ± 724
**8**	SP	ND	ND	ND	3.24 ± 0.43

^a^ Displacement of [^3^H]DAMGO, [^3^H]deltorphin-2, and [^3^H]U-69593 from membranes of CHO cells transfected with the human opioid receptors MOR, DOR, and KOR. ^b.^ Displacement of [^3^H][Sar^9^, Met(O_2_)^11^]SP from membranes of CHO cells transfected with the human NK1 receptor. All values are expressed as mean ± SEM, *n* ≥ 3. ND-not determined.

**Table 2 molecules-24-04460-t002:** Agonist potencies (pEC_50_) and efficacies (α) of analogs **1–7** determined on MOR, DOR, and KOR coupled with calcium signaling.

Peptide	MOR		DOR		KOR	
	pEC_50_ (CL_95%_)	α ± SEM	pEC_50_ (CL_95%_)	α ± SEM	pEC_50_ (CL_95%_)	α ± SEM
**EM-2**	8.22 (7.87–8.56)	1.00	Inactive		Inactive	
**DPDPE**	Inactive		7.29 (7.16–7.43)	1.00	Inactive	
**Dynorphin A**	6.67 (6.17–7.17)	0.83 ± 0.10	7.73 (7.46–8.00)	0.99 ± 0.04	8.86 (8.59–9.12)	1.0
**1**	8.98 (8.50–9.45)	0.98 ± 0.01	Crc incomplete		8.66 (8.56–8.76)	0.96 ± 0.02
**2**	7.50 (7.28–7.71)	0.66 ± 0.04	Crc incomplete		8.01 (7.56–8.46)	0.99 ± 0.05
**3**	6.76 (6.46–7.06)	0.72 ± 0.04	Crc incomplete		8.45 (7.56–9.34)	1.05 ± 0.05
**4**	7.46 (7.26–8.00)	0.90 ± 0.04	Crc incomplete		7.85 (7.60–8.11)	0.92 ± 0.05
**5**	7.63 (7.12–7.80)	0.86 ± 0.03	6.41 (5.82–7.01)	0.50 ± 0.01	7.17 (7.02–7.31)	0.64 ± 0.04
**6**	8.04 (7.89–8.20)	0.81 ± 0.05	6.82 (5.97–7.66)	0.65 ± 0.05	8.33 (7.96–8.69)	0.92 ± 0.02
**7**	8.69 (8.23–9.16)	0.85 ± 0.05	6.44 (5.80–7.08)	0.38 ± 0.04	8.80 (8.46–9.14)	1.06 ± 0.06

“Crc incomplete” means that the maximal effect could not be determined due to the low potency of a compound; endomorphin-2 (EM-2), DPDPE, and dynorphin A were used as reference agonists for calculating intrinsic activity at MOR, DOR, and KOR, respectively. Data are expressed as mean ± SEM, *n* = 5.

**Table 3 molecules-24-04460-t003:** Agonist potencies (pEC_50_) and efficacies (α) of SP and analogs **2**–**7** determined on NK1 coupled with calcium signaling.

No.	NK1	
	pEC_50_ (CL_95%_)	α ± SEM
**SP**	9.08 (8.83–9.34)	1.00
**2**	Inactive	
**3**	Inactive	
**4**	8.45 (8.00–8.91)	0.96 ± 0.05
**5**	7.80 (7.45–8.15)	1.11 ± 0.05
**6**	6.11 (5.73–6.49)	1.03 ± 0.05
**7**	5.68 (5.35–6.01)	0.99 ± 0.08

SP was used as a reference agonist for calculating intrinsic activity at NK1. *N* = 5.

**Table 4 molecules-24-04460-t004:** Antagonist potencies (pK_B_) of aprepitant and analogs **2** and **3**.

No.	pK_B_ (CL_95%_)
aprepitant	10.11 (9.48–10.74)
**2**	7.33 (7.03–7.63)
**3**	7.63 (7.25–8.00)

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
