# Peer review of "Synthesis and Pharmacological Evaluation of Hybrids Targeting Opioid and Neurokinin Receptors"

_molecules, 2019, doi:10.3390/molecules24244460_

Round 1

Reviewer 1 Report

The manuscript under consideration is quite an interesting one. The authors have given a very good introduction as a basis for their research. However, I have the following suggestions for the authors.

 1. The authors should explain why they choose the writhing test over other tests. Nowadays, it is less commonly used as a primary screening test for potential antinociceptive compounds. Formalin test could be a good choice for inflammatory pain. For testing opioid compounds writhing test is not the right choice of the experiment. Hot plate and tail-flick test should be included to understand the antinociceptive profile of the test compounds.

2. There must be a positive control to which the effect of the test compound needs to be compared rather than to the vehicle only. The author should explain why the omitted the positive control in their experiments. I would like to suggest the authors to include proper positive controls in the animal studies. This is required for the comparison of the observed effects with positive control.

3. The authors reported a lack of tolerance in the writhing test. Again, tolerance of opioids is better understood in tail immersion, hot plate test alongside inflammatory pain model. I think, only lack of writhes in the test is insufficient for commenting on tolerance of the opioid analogues.

4. What is the basis of choosing the tested dose of the compounds in the writhing test?

5. Only saline or the peptides were administered for understanding the impact on stool output. There was no opioid positive control to compare the effect. Comparing the effect with the saline is not sufficient as evidence.

6. The author should include the enzyme stability data of the compounds.

Reviewer 2 Report

The original paper by  Wtorek and colleagues aims at understanding the role of tachychinin  system in opioid-induced nociception. The paper is clearly written; hypothesis, objectives, methods, and results  have been well presented; discussion of the findings is also clear. The article is quite innovative and I find it acceptable in the present form.

Round 2

Reviewer 1 Report

Accept